# Gabor Filter-Based Segmentation of Railroad Radargrams for Improved Rail Track Condition Assessment: Preliminary Studies and Future Perspectives

**Gerald Zauner** [1,*] **, David Groessbacher** [2] **, Martin Buerger** [2] **, Florian Auer** [2] **and Giuseppe Staccone** [3]

1 School of Engineering, University of Applied Sciences Upper Austria, 4600 Wels, Austria
2 Plasser & Theurer GmbH, 4021 Linz, Austria; david.groessbacher@plassertheurer.com (D.G.); martin.buerger@plassertheurer.com (M.B.); florian.auer@plassertheurer.com (F.A.)
3 Ground Control Geophysik & Consulting GmbH, 82152 Planegg, Germany; georadar@saferailsystem.com
* Correspondence: gerald.zauner@fh-wels.at

**Abstract:** Ground penetrating radar (GPR) has been used for several years as a non-contact and non-destructive measurement method for rail track analysis with the aim of recording the condition of ballast and substructures. As the recorded data sets typically cover a distance of many kilometers, the evaluation of these data involves considerable effort and costs. For this reason, there is an increasing need for automated support in the evaluation of GPR measurement data. This paper presents an image segmentation pipeline based on 2D Gabor filter texture analysis, which can assist users in GPR data-based track condition assessment. Gabor filtering is used to transform a radargram image (or B-scan) into a high-dimensional, multi-resolution representation. Principal component analysis (PCA) is then applied to reduce the data content to three characteristic dimensions (namely amplitude, frequency, and local scattering) to finally obtain a segmented radargram image representing different classes of relevant image structures. From these results, quantitative measures can be derived that allow experts an improved condition assessment of the rail track.

**Keywords:** ground penetrating radar (GPR); railway; track condition assessment; image processing; 2D Gabor filter; image segmentation

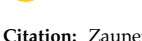

## 1. Introduction

Data acquisition by means of ground penetrating radar (GPR) in railway track maintenance is an established, non-destructive measurement method for monitoring ballast conditions and other parameters [1–4]. GPR can provide measurements of the layer thickness and composition (lateral and longitudinal changes) of individual subgrade layers, moisture content and contamination condition of the ballast. It can map the lateral and longitudinal changes of ballast and subgrade layers to provide information on failure and deformation conditions with minimal disruption to train operations. In this respect, GPR can be used as an effective tool to uncover the causes of recurrent poor rail track geometries and unstable roadways [5]. It is also possible to identify and better investigate problematic railway sections by considering, for example, successive inspection campaigns that show a systematic increase in geometric fault parameters over time [6].

For this purpose, the radar measurement system is usually mounted directly on a rail vehicle, as shown in Figure 1a. The unit of transmitter and receiver is called a transducer which emits pulses of radar waves (electromagnetic waves, usually with a center frequency of 200 MHz and above) towards the ground and receives their reflections (Figure 1b). During the radar survey of a railway track, radar pulse echoes (single reflection signals referred to as "traces" or A-scans) are continuously detected and recorded at regular intervals while the vehicle is moving forward (Figure 1c). In this way, a pictorial representation of the local radar reflection properties of the substructure can be generated

by combining the individually received reflection signals column by column to form a so-called radargram (or B-scan, Figure 2). This allows trained personnel to examine the subsurface structure of the track and to detect any anomalies (e.g., changes in the depth of the track planum, estimation of the degree of contaminations, and the moisture content as well as accumulated water and settlements, etc.). After manual evaluation of the recorded data, specific maintenance work can be derived. Thus, the evaluation (i.e., interpretation and categorization) of this measurement data requires a visual inspection, usually carried out by persons with geophysical training and experience. However, even for experienced persons familiar with railway radargrams, the accurate and reproducible labeling, e.g., of sub-track layer boundaries, still poses a practical challenge. On the one hand, GPR measurements can be carried out at high speeds of more than 100 km/h, so that large distances can be covered in a very short time. On the other hand, the subsequent manual evaluation of the collected image data is very time-consuming. Automatic support of this process could significantly reduce inspection times and increase objectivity in the evaluation.

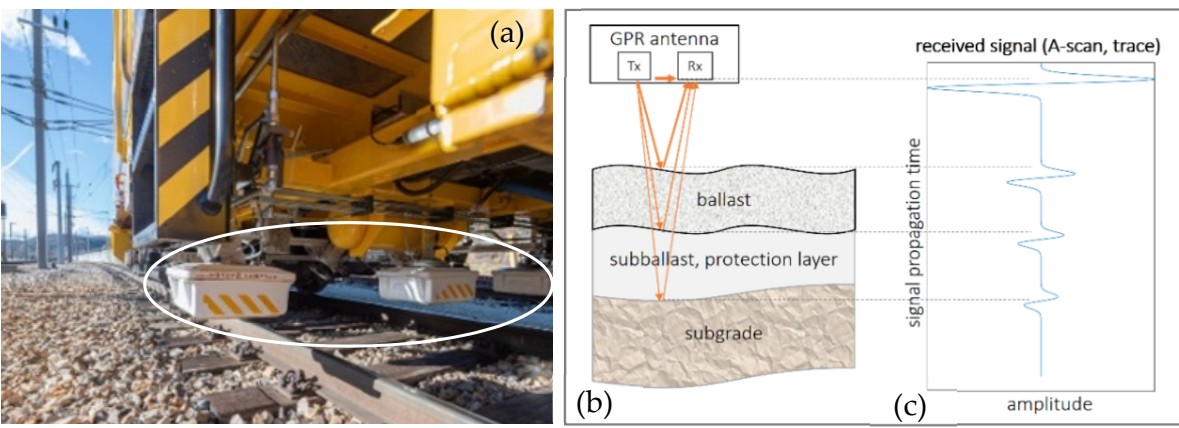

**Figure 1.** (**a**) Three GPR transducers mounted on a rail vehicle (indicated by the white ellipse), (**b**) principle of signal generation: typical subsurface structure of a railway track, and (**c**) the corresponding reflected radar signal amplitudes (A–scan or trace), which typically occur strongly at well–defined layer boundaries with different dielectric properties.

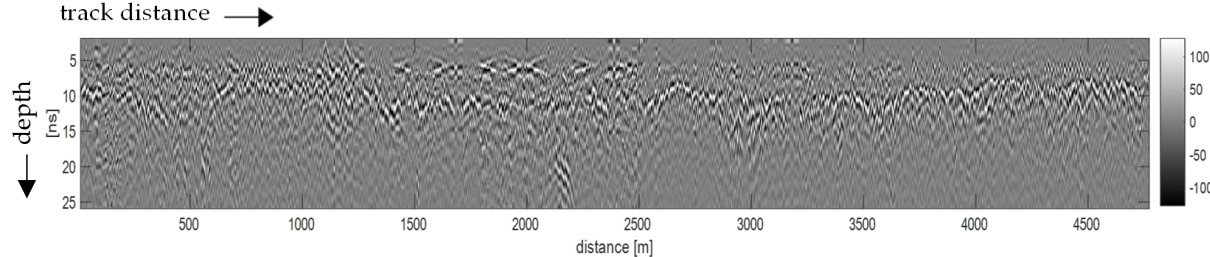

**Figure 2.** Typical radargram (or B–scan): the *x*-axis represents track distance (usually covering several kilometers), and the *y*-axis shows depth (typically indicated by radar signal propagation times).

Different approaches for automated support in the evaluation of GPR data in railway maintenance are reported in the literature. Frequently, these methods are based on time domain analysis or frequency domain analysis of individual GPR signals (traces) e.g., by means of 1D Fourier transform [7,8]. Combined time–frequency analysis methods are reported in [9–11] but also applied only to single signal traces (1D). Typical applications refer to the quantification of ballast conditions, geometric ballast properties, or the measurement of moisture content [12–16]. In [17,18], the application of image processing approaches supported by machine learning methods (image classification) is described to characterize mainly ballast conditions and ballast stone size distributions. Recent attempts deal with

deep learning approaches for image-based classification, e.g., object detection methods [19], but with the disadvantage that a large amount of high-quality labeled ground truth data is needed in advance to achieve good results.

The aim of the method presented here is to provide automated support for the exact (pixel-wise) segmentation of relevant image regions in typical railway radargrams. The data basis for our evaluations was provided by Groundcontrol GmbH, a company that has surveyed more than 100,000 km of railway tracks worldwide using GPR. With many years of experience in the interpretation of such measurement data, experts from this company have accompanied the development of the presented method. The novelty of this work consists in the application of 2D Gabor filtering for image texture analysis and the interpretation of the results of a subsequent dimensionality reduction by means of principal component analysis (PCA). It is shown that the first three principal components can be assigned to visually interpretable image structures, which are subsequently used for a pixel-precise segmentation of relevant image areas. Technically, the presented method can be classified as a two-dimensional combined time–frequency analysis approach for segmenting radargram images. As a result, the exact subdivision of radargram image structures becomes possible in a level of detail that far exceeds the purely visual capabilities of a human observer, e.g., for determining the exact course of ground structure boundaries.

The paper is structured as follows: Section 2 presents in detail the segmentation methodology based on image texture analysis using 2-dimensional Gabor filters and principal component analysis (PCA). In Section 3, the results of the proposed method are shown using typical radargrams from GPR railway surveys. In Section 4, a refinement of the proposed method is presented, which allows to additionally consider phase information in the segmentation process. Finally, a summary and outlook are given in Section 5.

## 2. Radargram Segmentation Method

### 2.1. Overview of the Proposed Methodology

An introductory overview of the presented segmentation method is shown in Figure 3. First, the raw radargrams go through the same well-defined pre-processing steps routinely applied in GPR signal analysis (e.g., gain correction, filtering, signal whitening, start signal correction, etc.) to ensure comparability between different recordings. The results are then available in the form of an 8- or 16-bit grayscale image. Then, a multiscale 2D Gabor filtering is applied, which is described in detail in the following Section 2.2. A subsequent dimensionality reduction using PCA finally reduces the high-dimensional data representation to three characteristic feature dimensions (see Section 2.3). With this reduced representation (based on the features "amplitude", "frequency", and "local scattering", as will be shown in Section 2.3), certain texture properties can then be precisely defined in a simple and objective manner. A corresponding pixel-based classification is then performed, which extracts exactly those regions in the radargram that match this texture definition. The image areas segmented in this way can then be further analyzed by means of image processing (e.g., edge detection).

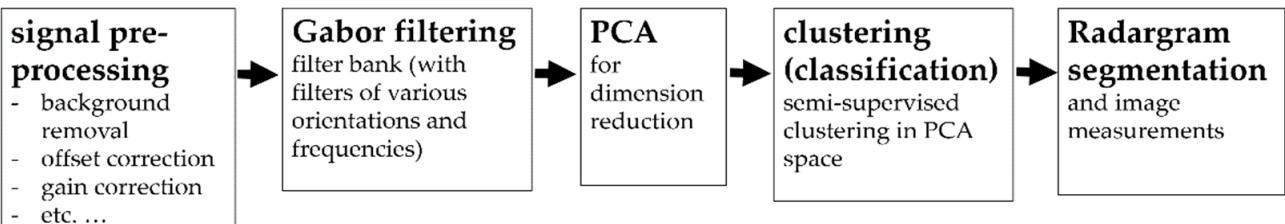

**Figure 3.** Overview of the proposed image processing chain for radargram segmentation.

### 2.2. Radar Image Texture Analysis Based on 2D Gabor Filters

The typical image content of railway radargrams is mainly characterized by local wave-shaped 2D structures with different spatial frequencies, orientations, and amplitudes. This motivates the use of so-called 2D Gabor filters (Figure 4), which are complex functions typically used for texture analysis in the field of digital image processing and which also can represent different spatial frequencies (and orientations) to quantitatively describe local image structures (textures).

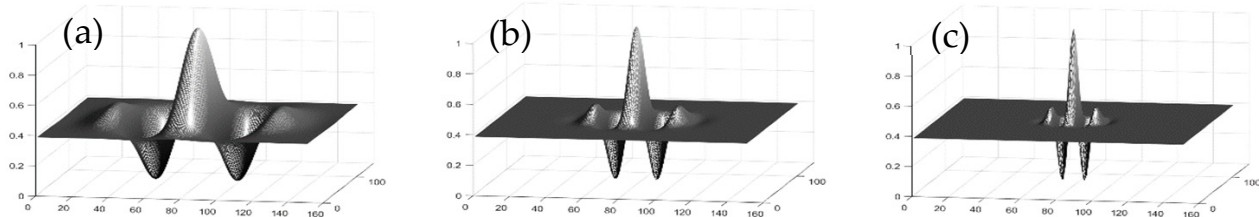

**Figure 4.** 3D representation of 2D Gabor filter kernels (real part representation) of various wavelengths (or spatial frequencies/scales) representing (**a**) low-frequency, (**b**) mid–frequency, and (**c**) high-frequency wave patterns. These filters can also be rotated.

Gabor filtering is a widely applied technique for texture analysis having optimal localization properties in both spatial and frequency domain, and they have been successfully used in many image analysis applications [20]. Frequency (or scale) and orientation are two key parameters of the Gabor filter, which detects the presence of a given spatial frequency content in an image in a given direction around a local pixel neighborhood, (i.e., Gabor filters compute one filtered value at each pixel position taking its spatial neighborhood into account). Additionally, they can be tuned to different scales and orientations to create a so-called filter bank (Figure 5).

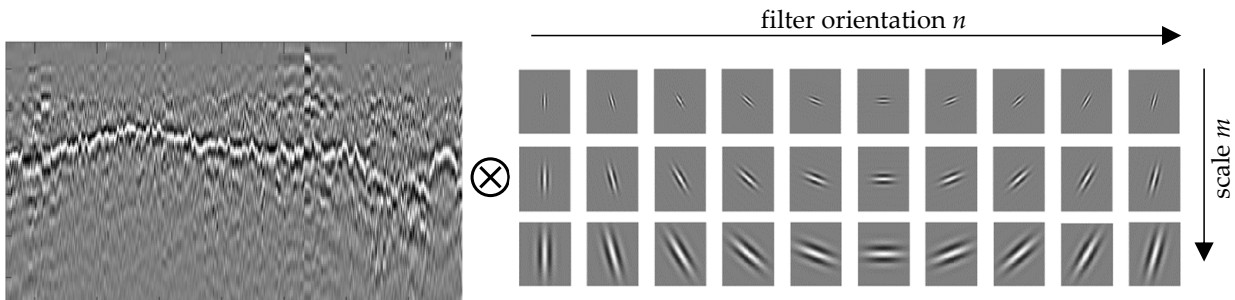

**Figure 5. Left**: typical radargram (B-scan) after standard signal pre-processing; **right**: some Gabor filters used for analysis (2D real part representation). These Gabor filter kernels can be changed in frequency (scale $m$) and in orientation (angle $n$) and then form a so-called filter bank.

For a radargram image $I(x, y)$ with size $p \times q$, its discrete Gabor transform is given by a convolution [21]:

$$G_{mn}(x,y) = \sum_{s=0}^{K} \sum_{t=0}^{K} I(x-s, y-t) g_{mn}^{*}(s,t) \tag{1}$$

where $K$ is the filter mask size and $g_{mn}^{*}$ is the complex conjugate of $g_{mn}$, which is a class of self-similar wavelets generated from dilation and rotation of the following mother wavelet:

$$g(x,y) = \frac{1}{2\pi\sigma_x\sigma_y} \exp\left(-\frac{1}{2}\left(\frac{x^2}{\sigma_x^2} + \frac{y^2}{\sigma_y^2}\right)\right) \cdot \exp(j2\pi f x) \tag{2}$$

where $f$ is the modulation frequency and $\sigma_x$, $\sigma_y$ represent the standard deviations in the $x$ and $y$ directions. The self-similar Gabor wavelets are obtained through the generating function

$$g_{mn}(x,y) = a^{-m}g(\widetilde{x},\widetilde{y}) \tag{3}$$

where $m$ and $n$ specify the scale and orientation of the wavelet, respectively, with $m = 0, 1, \dots, M - 1$, and $n = 0, 1, \dots, N - 1$, and

$$\widetilde{x} = a^{-m}(x\cos\theta + y\sin\theta) \tag{4}$$

$$\widetilde{y} = a^{-m}(-x\sin\theta + y\cos\theta) \tag{5}$$

where $a > 1$ and $\theta = n\pi/N$.

After applying Gabor filters with different orientations $n$ and different scales $m$, a set of magnitudes E at each pixel position is obtained:

$$E(x,y,m) = \sum_n |G_{mn}(x,y)| \tag{6}$$

where $m$ is the wavelet scale and $n$ is the wavelet orientation.

Usually, the sum is calculated over all orientations $n$ (from 0° to 180°) to finally have $m$ different filter response maps (according to the number of scales) representing different frequency content, i.e., the generated magnitude maps pixel-wise represent the spatial signal energy distribution at every scale. For more specific applications, the number of angles could also be selectively adapted.

Figure 6 exemplarily shows some of the filtered results and how different image features are captured by the Gabor filters. Those image areas that structurally correspond to the Gabor filter in use will show accordingly high filter responses, i.e., high-frequency information is captured at lower scales and as scale increases, less detailed low-frequency signal content is emphasized (so, scale and spatial frequency are inversely related). As mentioned above, for a number $m$ of different filter scales, the corresponding $m$ filter responses are determined for each pixel position. As an example, the first three resolution levels of a radargram are shown in Figure 6b–d. Each pixel in the radargram can then be represented by a $m$-dimensional vector $v_{x,y}$ or accordingly by a single data point in a $m$-dimensional feature space (Figure 6a). In general, similar local image structures (textures) in this feature space representation will form local clusters with a small Euclidean distance to each other—i.e., conversely, image structures with significantly different appearances (in terms of frequency and amplitude) will then have a feature space representation at a correspondingly larger distance and thus be easily separable from each other.

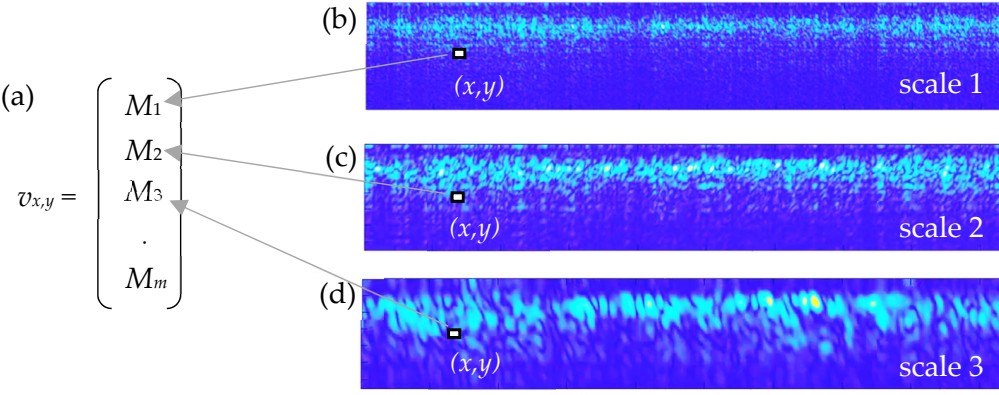

**Figure 6.** Pixel-wise definition of (**a**) a feature vector $v_{x,y}$ from Gabor filter maps at various resolution levels (**b**–**d**), representing different frequency content from high-frequency ($M_1$) to successively lower frequency content ($M_2, M_3, \dots, M_m$). Colors indicate different filter response magnitudes.

### 2.3. PCA of Gabor Filtered Radargrams

Subsequently, the dimensionality of this feature space representation is reduced using PCA (also known as discrete Karhunen–Loève transform) with the goal of segmenting the radargram image into different texture classes, each with a visually characteristic "appearance". PCA represents a well-established mathematical procedure with the help of which the dimension of a feature space can be reduced without significant loss of information [22]. This means that highly correlated signals/attributes have only a low information content and the benefit, for example, a classification task is lower than that of attributes with low correlation. The goal of PCA is to de-correlate the attributes by an orthogonal basis transformation. In this way, the underlying coordinate system is rotated in the direction of the main axes, which run in the direction of the largest variance of the data points. The working principle of PCA is to look for the projection with the smallest mean-squared distance between the original data and their projections on to the new (rotated) basis system, which turns out to be equivalent to maximizing the variance of the data points. Principal components refer to the new variables constructed as a linear combination of the initial features (i.e., Gabor filter responses $M$ in our case), such that these new variables are uncorrelated. Since the principal components are linearly independent, they are orthogonal to each other in a Cartesian coordinate system.

Based on the Gabor feature set matrix F (with size $p \cdot q \times m$, where $p \cdot q$ is the number of total pixels in the radargram and $m$ the number of Gabor filter scales, Equation (7)), PCA returns the $m$ principal component coefficients (representing the directions of the axes with most variance):

$$F = \begin{pmatrix} M_{1,1} & \cdots & M_{1,m} \\ \vdots & \ddots & \vdots \\ M_{pq,1} & \cdots & M_{pq,m} \end{pmatrix} \tag{7}$$

The actual dimensionality reduction is performed after this transformation by selecting only those principal components that meet a certain minimum variance requirement (i.e., that can explain a large amount of the total data variance). PCA compresses as much information as possible into the first principal component, the remaining into the second, and so on. Our texture analysis experiments have shown that already the first three principal components typically can explain about 97% of the signal variance in our radargrams (representing second order statistics).

Usually, these new variables do not have an interpretable meaning, being a linear combination of abstract features. In connection with the analysis of radargrams, however, these first three principal component axes can indeed be assigned a specific (visually interpretable) meaning. In this way, the first component describes the amplitude of the signal, the second the frequency, and the third is referred to as the "local scattering" property. While the first two properties have an unambiguous interpretation, the third one needs to be explained in more detail. An example is shown in Figure 7, where the top image shows coherent low-frequency radargram structures without "local scattering" and the example below shows structures with the same frequency content, but with superimposed high-frequency (most likely local wave scattering induced) noise-like patterns.

After projection of the original feature space on to the new 3D coordinate system (with the according basis axes representing amplitude, frequency, and scattering), the distribution of all radargram pixels can be visualized in the form of a 3D scatterplot (Figure 8a). In addition, the corresponding 2D projections are shown in Figure 8b–d. Specific parts of this 3D feature space now represent distinct radar signal properties, which in turn can be assigned, e.g., to characteristic radargram features (under the supervision of experts with experience in the interpretation of track radar signals). By simply defining a particular "region" in the 3D feature space (i.e., characterizing it in terms of amplitude, frequency, and scattering), the corresponding pixels (i.e., spatial position) in the radargram can be identified accordingly. This allows the expert the individual (and now objectifiable) definition of any number of specific classes in the 3D feature space (scatterplot coordinates), which,

according to individual expertise, represent important subsurface structures. The black dots in Figure 8 exemplarily indicate the position of possible cluster centers that can be used for classification (i.e., the Euclidean distance to these centers determines the corresponding classification/coloring of the data points). The according visualization of these datapoints in the radargram results in a pixel-wise segmented image that automatically indicates characteristic rail track substructures, where the color indicates the corresponding class.

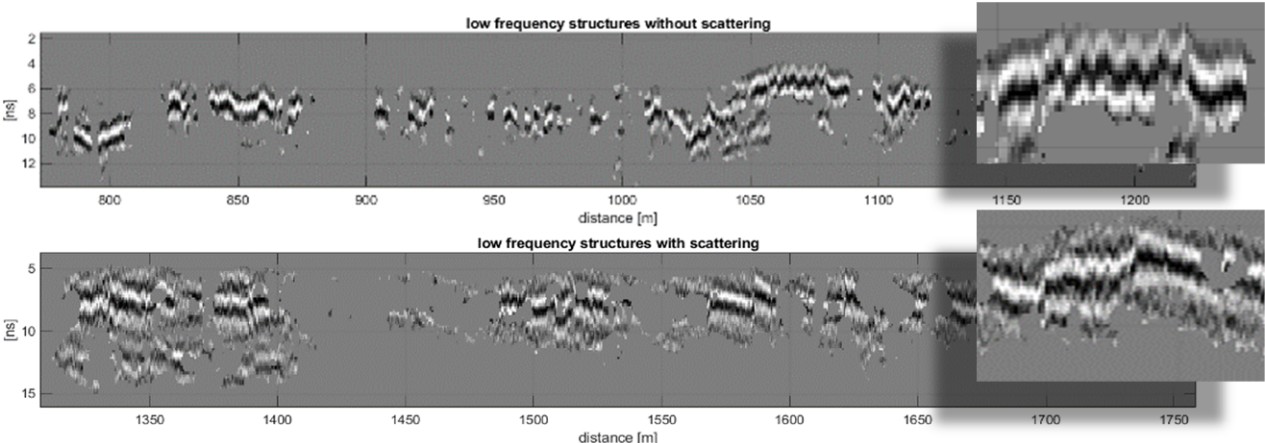

**Figure 7. Upper**: typical low-frequency structures without local scattering; **lower**: low-frequency structures with strong local scattering, which manifests as an overlaid noise-like pattern.

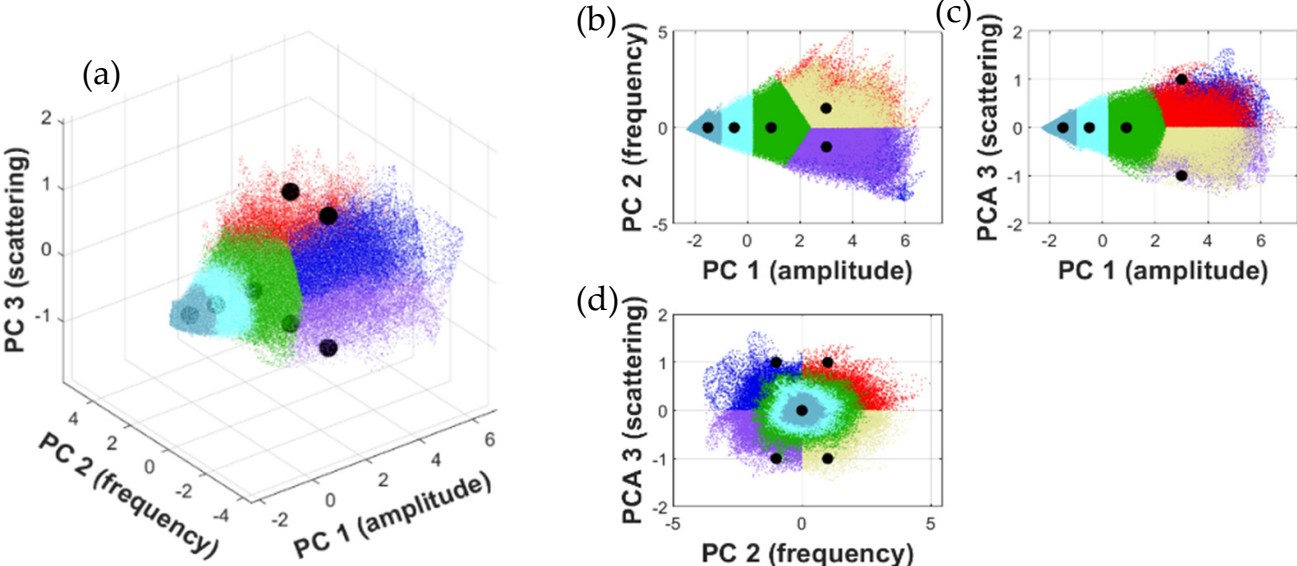

**Figure 8.** (**a**) Typical 3D feature space representation (3D scatterplot), where each colored data point represents a spatial pixel in the original radargram. The 2D scatterplots on the right show the same information but as (**b**) horizontal, (**c**) vertical, and (**d**) lateral projections.

## 3. Automatic Radargram Segmentation Results

Typical results of the segmentation procedure are presented in this chapter. The examples shown are based on radargrams that were recorded with a 400 MHz GPR system at measurement speeds of approx. 100 km/h with sampling rates of 1024 samples/trace. If, for example, the task is to identify very low-frequency content ranging from specific medium- to high-amplitude values (which typically represents the so-called track planum, the boundary between the ballast and sub-ballast layer), simply the lower right part of

the 3D feature space in Figure 8a must be selected and the corresponding pixels (image areas) can immediately be displayed in the segmented radargram (exemplarily indicated as blue/purple regions in Figure 9b). These dominant structures can also be recognized quite well visually in the raw radargram (Figure 9a).

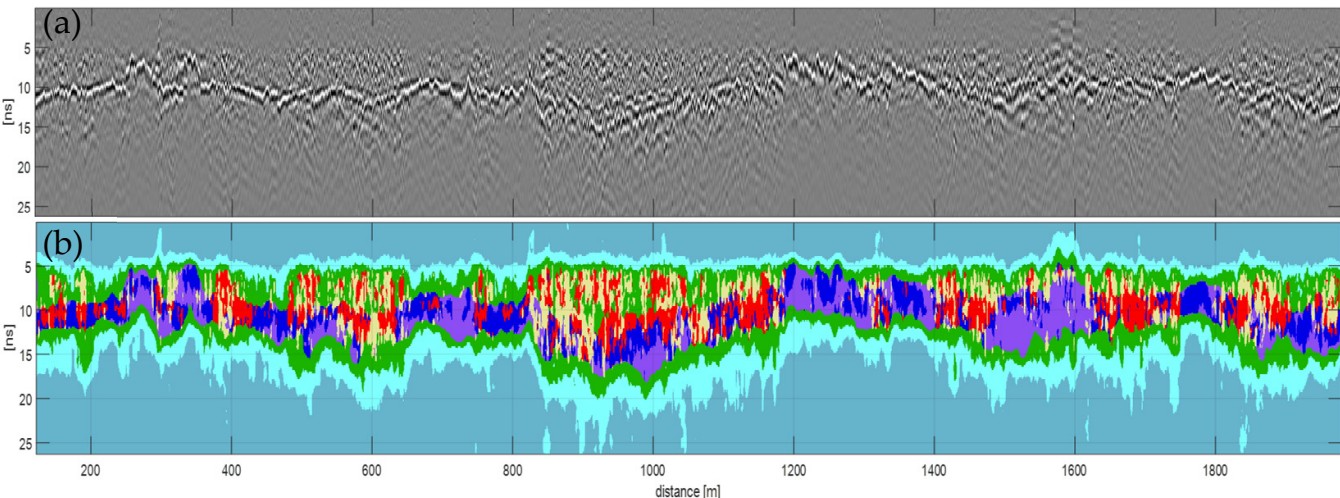

**Figure 9.** (**a**) Original radargram; (**b**) segmentation result of the proposed method, where different colors represent different relevant radargram structures based on objectifiable criteria (scatterplot coordinates) defined by railway radargram experts (e.g., blue/purple regions: track planum; yellow/red regions: indications of ballast fouling).

High-frequency image structures with pronounced local scattering, on the other hand, typically indicates ballast fouling (i.e., when the finer materials mix with fresh ballasts due to heavy repeated train loads), which is shown as yellow/red areas in the segmentation result (Figure 9b). These structures are much less pronounced and more difficult to detect visually in the original radargram.

Another possible representation is shown in Figure 10, where five automatically extracted image segments of an examined railroad section are visualized separately. Each segment is characterized by specific signal frequency and amplitude ranges. This allows further investigations to be carried out, such as determining the signal energy or measuring the area (or "thickness") of these specifically characterized texture segments, in order to automatically monitor the exceeding of critical threshold values (e.g., the amount of locally collected water typically represented by low-frequency and very high-amplitude regions without local scattering contributions).

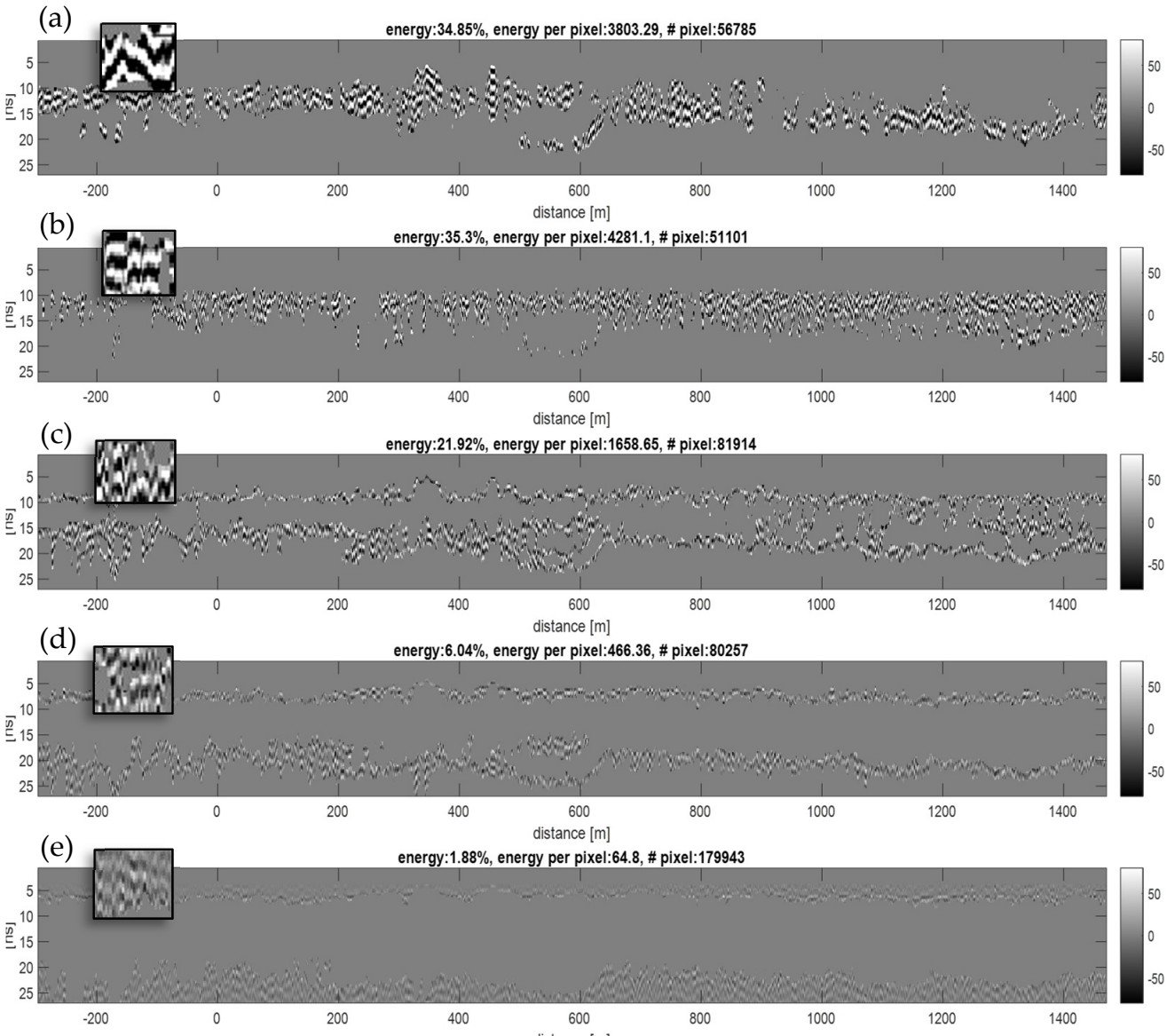

**Figure 10.** Five examples of automatically extracted segments of a radargram characterized by typical textures: (**a**) low-frequency and high-amplitude regions, (**b**) medium-/high-frequency and high-amplitude regions, (**c**) medium frequency and medium amplitudes, and (**d**) medium- and (**e**) low-amplitude regions.

Figure 11a exemplifies another advantage of this special form of automatic radargram segmentation. Due to the complex image structure in the raw images, it is difficult even for the trained human eye to determine the exact course and delineation of different relevant textures. The proposed automatic segmentation offers advantages here since, e.g., the exact course of the layer boundaries is now based on comprehensible criteria and no longer on a purely visual (subjective) assessment, which could possibly lead to different results for different assessors. Various further types of possible automated evaluations are shown in Figure 11b–d, where, e.g., upper and lower formation boundaries have been extracted automatically. Subsequently, the exact course of these contours is the basis for further evaluations, for example to determine any undulations in the boundary layers or to determine the exact depth and thickness of substructure layers. In this way, potential problem areas can be identified very quickly in an automated way, which provide useful indications for track maintenance.

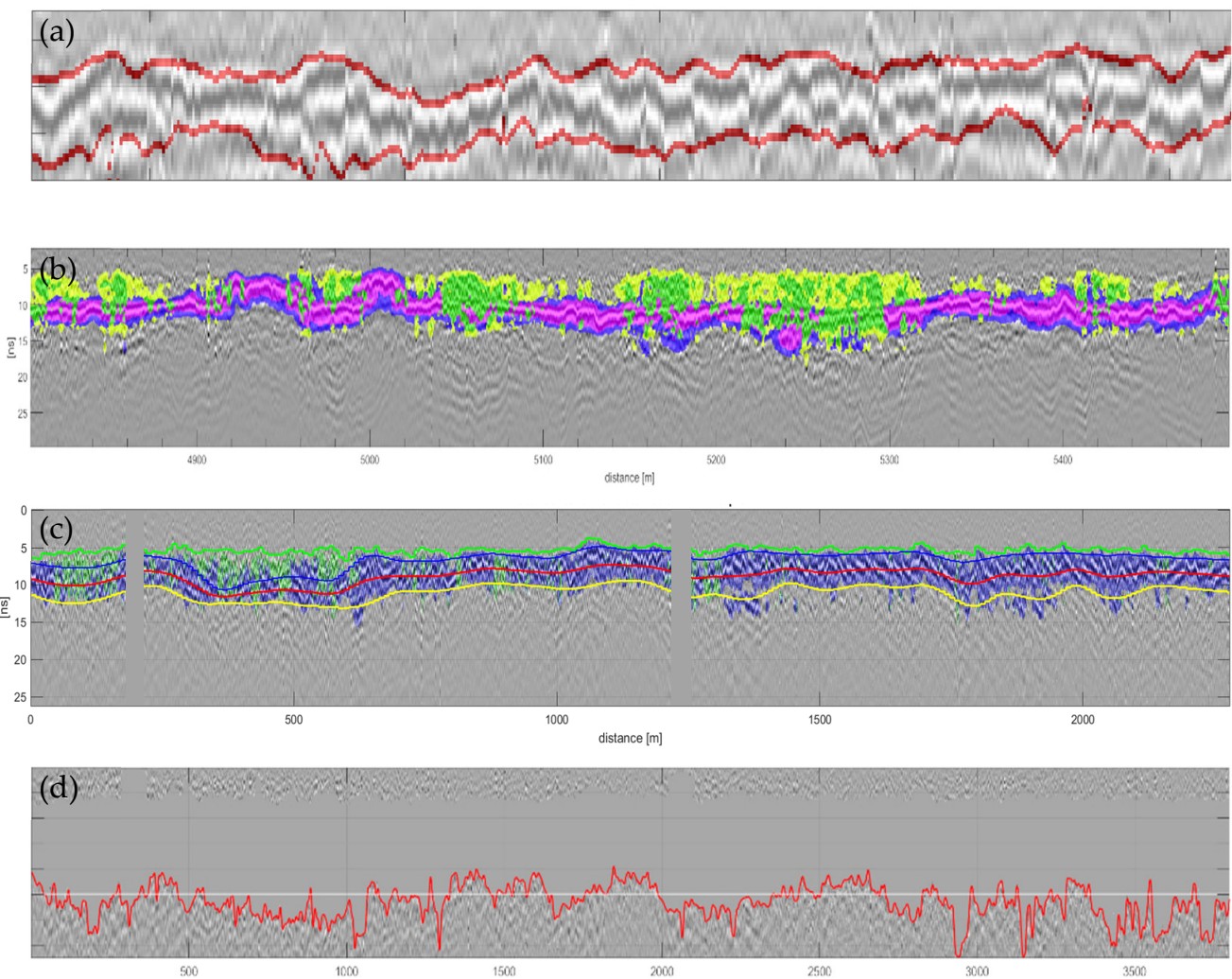

**Figure 11.** (**a**) Automatically extracted upper and lower limits of a railway planum (indicated in red), (**b**) a segmentation example where different colors indicate high– and low-frequency segments having high amplitudes exceeding a certain threshold; (**c**) automatic layer boundary extraction (green, red, yellow) relevant for water content estimation; (**d**) automated line extraction (red) for subgrade analysis.

## 4. Phase Measurements at Low-Frequency Layer Interfaces

The segmentation process shown above can also be extended to include signal phase evaluation. Generally, the phase of the reflected GPR signal is influenced by the wave impedance. It is negative when the signal is split at an interface where the impedance decreases (i.e., the relative permittivity increases). Conversely, the phase is positive when the signals are split at an interface where the velocities and wave impedances increase (i.e., the relative permittivity decreases). Thus, if the relative permittivity of a target is lower than that of the surrounding medium (e.g., air-filled void or plastic pipe), there is no phase reversal of the backscattered GPR wave—on the contrary, e.g., especially for water, a phase reversal is produced. In this way, additional information about the boundary layer conditions (e.g., water content) or about possibly buried objects (especially of metallic ones) could be derived.

Based on the segmentation process described in this paper, phase shifts in extracted very low-frequency segments of the radargram can be determined. Signal traces $f^{trace}$ with a specific phase angle are identified by applying a 1D cross-correlation (CC) with Ricker wavelets $g_\theta^{Ricker}$ of tunable phase from $-180°$ to $170°$ in steps of $5°$ (Equation (8), Figure 12). The Ricker wavelet (also referred to as Mexican Hat wavelet) is used here

because of its close similarity to the measured radar reflection signal. In other words, cross-correlation is used to determine that Ricker wavelet that best matches the shape of the current signal trace:

$$\mathrm{CC}(\theta) = \operatorname*{argmax}_{\theta}((\mathrm{f}^{\mathrm{trace}} * \mathrm{g}_{\theta}^{\mathrm{Ricker}})(\tau)) = \operatorname*{argmax}_{\theta}\left(\int \mathrm{f}(t)\mathrm{g}_{\theta}(t+\tau)dt\right) \qquad (8)$$

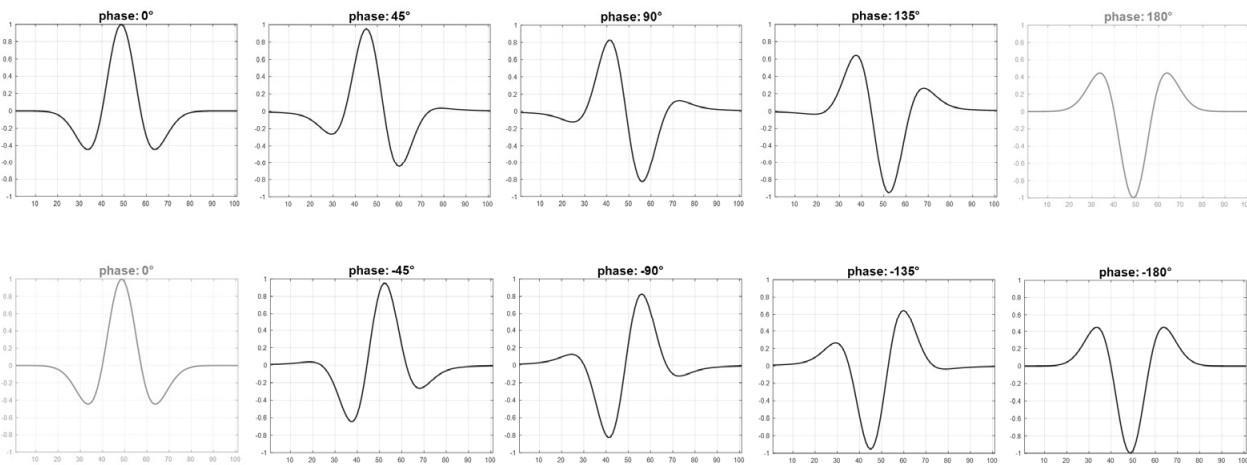

**Figure 12.** Ricker wavelets with different phase angles.

This additional signal phase information can be used for advanced classification and segmentation tasks where the identification of phase angle changes might be a valuable additional feature (e.g., for water content analysis or detection of buried objects). Figure 13a shows exemplarily an extracted low-frequency segment. Then a further subdivision of this segment is performed by defining a phase angle threshold as a distinguishing feature: those traces with a phase angle equal to approx. 0° are shown in the middle image, (Figure 13b) and those with a phase angle equal to approx. 180° are shown in the lower image (Figure 13c). As can also be perceived visually, the middle image consists of signals with a phase angle of approx. 0°, i.e., the signal maximum (bright) is clearly recognizable here in the central area of the image segment. In the lower image, on the other hand, the central peak area is continuously pronounced as a signal minimum (dark) indicating phase angles of 180°.

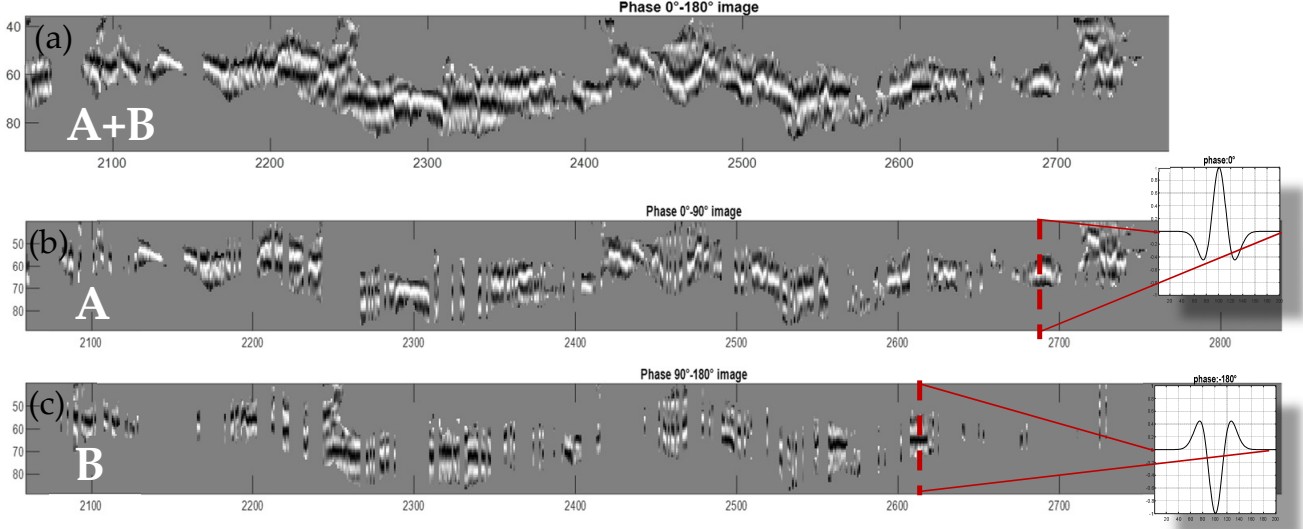

**Figure 13.** The extracted low-frequency radargram segment (segment (**a**)) can be further segmented according to the phase angle of the individual radargram traces (segment (**b**) and segment (**c**)).

## 5. Summary and Conclusions

The presented form of automatic texture segmentation based on 2D Gabor filtering contributes to reducing the complexity of interpreting and quantifying radargrams in rail track condition assessment. Based on the presented approach, it is immediately recognizable whether, for example, anomalous characteristics of radargram structures are present. The algorithm identifies those image areas that appear similar in terms of spatial frequency, amplitude, and "local scattering"—the individual interpretation of these characteristic image structures still must be carried out by experts. However, it can support the evaluation process and significantly increase the reproducibility of results, since the same objective analysis criteria are applied to assess characteristic image structures which can then be analyzed in detail, e.g., to support the difficult process of determining critical track conditions such as ballast fouling or clay fouling. This process is accompanied by automated image measurements, such as the identification of different layer structures and the corresponding measurement of layer thicknesses and layer depths. Even the extraction of signal phase information at layer boundaries becomes feasible. In this way, typical radargram sections covering areas of several kilometers of track can be analyzed on modern computers within a few seconds. The radargram analysis presented is already routinely used in track assessment by Groundcontrol GmbH as a supporting method, and the degree of automation has already been significantly improved in this way.

The possibilities of pixel-precise segmentation will be further explored in future activities. This includes, for example, the detailed comparison with data from bore holes or trial pits. Future work should additionally show whether the segmentation proposed here can also be used for automated labeling in conjunction with modern deep learning methods (i.e., to efficiently generate learning data for supervised learning algorithms in this way), which could then in turn be used to further improve the automated interpretation of radargrams. The presented approach thus represents a further step towards advancing the reproducibility and objectifiability of GPR analysis of track substructures and thus increasing the safety of railways. However, the presented approach can presumably also be transferred to other areas of application in which automated radar signal evaluation is of importance.

**Author Contributions:** Conceptualization, G.Z.; methodology, G.Z.; software, G.Z.; validation, D.G., G.S.; formal analysis, G.Z.; investigation, G.Z., D.G., G.S.; resources, M.B., F.A., G.S.; data curation, G.S.; writing—original draft preparation, G.Z.; writing—review and editing, D.G.; visualization, G.Z.; supervision, G.S.; project administration, M.B., F.A. All authors have read and agreed to the published version of the manuscript.

**Funding:** This research received no external funding.

**Institutional Review Board Statement:** Not applicable.

**Informed Consent Statement:** Not applicable.

**Data Availability Statement:** No data available.

**Conflicts of Interest:** The authors declare no conflict of interest. The funders had no role in the design of the study; in the collection, analyses, or interpretation of data; in the writing of the manuscript; or in the decision to publish the results.

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
