# Peer review of "Gabor Filter-Based Segmentation of Railroad Radargrams for Improved Rail Track Condition Assessment: Preliminary Studies and Future Perspectives"

_remotesensing, doi:10.3390/rs13214293_

Round 1
Reviewer 1 Report
The predictive maintenance of the railway infrastructures is a key action for any public authorities and private companies operating in the transportation and mobility sector. To date, there is a growing interest for novel applications of no-invasive, friendly-user and cost-effective electromagnetic sensing technologies for civil infrastructure monitoring and surveillance.
Then, a paper concerning the use of the GPR technology for the rapid mapping of ballast and near surface structures along the railway tracks is welcomed and must be encouraged.
However, I have some critical comments to the manuscript.
The mathematical background of the Gabor filter and PCA method is clearly presented and described, but the interpretation of the results is quite ambiguous. What are the limits and the advantages of this approach in experimental tests? What are the real cases that could be solved using this approach? More details about this key question is mandatory.
It quite ambiguous the sentence "under the supervision of experts with experience in the interpretation of track radar imaging". The authors must specify if the procedure is supervised or unsupervised. In many parts of the manuscript they claim the complete automation of the procedure.
The paragraph 6 concerning the phase of the GPR reflected signals strongly requires an improvement, the mathematical background and the interpretation of the results could be better described.
A comparison with other approaches is completely absent in the paragra Discussion.
It could be interesting to give some information about the execution and processing time in computer performance for GPR analysis and the possibility to apply the procedure in real-time.
Considering that the overall quality of manuscript is quite good, I suggest the pubblication of this paper after an accurate minor revision.
Reviewer 2 Report
This paper does not present any ground truth and hence it is impossible to determine whether the proposed approach has any validity. Until the authors can prove the results by reference to an adequate level of statistical sampling the approach is no more than speculation.
Reviewer 3 Report
Changes needed for paper "Gabor-Filter Based Segmentation of Railroad Radargrams for Improved Rail Track Condition Assessment"
Individual Changes;
line 29 -- [1]-[6] change to [1-6] ....
others:
(Line 77) [7], [8] to [7,8]
(Line 78) [9], [11] to [9-11]
(line 80) [12]-[16] to [12-16]; [17],[18] to [17,18]
Line 38 -- change Fig. 2 to Figure 2 ;
and everywhere in the document -- Line 103, 106, 126, 162, 240, 246, 279, 284, 291, 352, 412 and 425;
On Line 352 change Fig. 10b-c to Figure 10b,c ....
One can link the numbered Figure in text to the Figure indicated.
Line 103 -- 8- or 16-bit to 8 or 16-bit ....
Line 130 -- numbered equation change number to right margin of text (a little farther to the right than it is);
Do this for all numbered equations. Line 135, 140, 145, 146, 225, 414.
Line 221 -- change Eq. 7 to Equation 7 ....
Lines 477 - 520 -- References, list page numbers of all articles.
example: Line 477 Reference 1 - with page number of article included
1 I.L. Al-Qadi, W. Xie, and R. Roberts, "Scattering analysis of ground-penetrating radar data to quantify railroad ballast
contamination, " NDT & International, vol. 41, no. 6, Sep. 2008, 441-447, doi: 10.1016/j.ndteint.2008.03.004
if specific chapters of referenced books are used note the page numbers also; otherwise just the book information is just fine.
When listing several authors, MDPI tends to remove the “and” in the list.
Example:
I.L. Al-Qadi, W. Xie and R. Roberts changes to I.L. Al-Qadi, W. Xie, R. Roberts
MDPI reference list and formatting style guide link -- https://www.mdpi.com/authors/references
Paper structure changes:
In the INTRODUCTION one must indicate the goal of the study, proposed work to be shown, etc. In the final paragraph one must describe what is to be discussed in each of the remaining sections of the paper.
Example:
In this paper we are proposing a method to support evaluation of GPR measurement data using Gabor-filter based segmentation of Railroad Radargrams. In Section 2, we discuss …. In Section 3, we describe ….In Section 4,…In Section 5…In Section 6, …Summary and conclusions are discussed in Section 7.
Develop your own wording. The above is just an example. You might look at combining parts of Section 2 into the Introduction to achieve this.
Review the MDPI Author Layout Guide -- Link -- https://www.mdpi.com/authors/layout

Reviewer 4 Report
GENERAL OVERVIEW
The paper deals with the issue of GPR-based rail track condition assessment. The multi-step algorithm for signal processing is introduced. Some interesting results obtained using 2D Gabor filtering and principal component analysis (PCA) are presented and discussed.
The undertaken problem is of great interest of NDT researchers and it meets the scope of the journal. During review, no substantive problems were identified, however, the submission needs some improvement. In my opinion, the paper can be published in Remote Sensing after a major revision, on the condition that Authors make an effort to enhance the manuscript, especially regarding the way of presenting the results. The particular critical remarks are presented below.
CRITICAL REMARKS
1) Type of paper. The submission was categorized as ‘Communication’ whereas the in the manuscript the type of paper is ‘Article’, which has more strict requirements. A disambiguation is needed in this aspect.
2) Structure. The structure is logical, the following sections are organised reasonably. However, if possible, I encourage to reorganise the article using standard style (Introduction, Materials and Methods, Results and Discussion, Conclusions). The combination of description of applied methods with results makes the article less clear than in the IMRaD structure.
3) Formatting. The Journal’s template is generally followed, however, some derogations were detected, like redundant indents after equations (e.g., lines 137 and 142-143, 148, 155) The manuscript should be reviewed carefully according to the Journal’s template. Additionally, some unnaturally short paragraphs should be merged, e.g., lines 157-166 or 202-223; single sentence from line 389 should be attached to the previous paragraph in line 358.
4) Language. In general, the quality of the language used is sufficient, a few minor changes are required. I strongly encourage Authors to perform a careful proofreading of the whole text to eliminate typographic errors (e.g., two double space before ‘estimation’ in line 40, lack of space in ‘(Fig.7)’in line 279, lack of delimiter before and after ‘i.e.’ or ‘e.g.’ throughout the whole manuscript) and language inconsistencies (e.g., repeated information in line 72: ’very time-consuming and takes a long time’ or wrong verb form in line 81 – ‘is’ instead of ‘are’, it deals with ‘application’). Some sentences are complex making them difficult to follow and should be divided into subclauses (e.g., single sentences in lines 95-98, 105-109 or 121-125).
5) Introduction. The Introduction section is very short. It would be beneficial to include some general information about GPR applications and to discuss the main advances in the topic of GPR-based rail condition assessment in detail.
6) Novelty. I cannot find any direct information about the original contribution of the current paper. It should be formulated in the view of the current state-of-the-art and put at the end of the Introduction section.
7) Abbreviations. In the case of PCA, it would be better to put the full expansion before abbreviation (section title, line 201; and the text, line 203). Moreover, if PCA is expanded in line 91, the further use of only abbreviation is acknowledged.
8) Equations. The font should be changed to Palatino Linotype (some equations are written with different font). The equations should be centred and the equation numbers should be aligned right. The italic variables in equations should also be italic while mentioned in the main text – the rule is sometimes not respected, e.g., ‘m’ and ‘n’ (line 142) or ‘M’ (line 217). What is more, multiplication should not be a full stop (‘.’) but an interpunct (‘·’) – corrections needed in lines 135 and 220. Additionally, what do sigmas in eq. (2) denote? Please, explain.
9) Figures. The figures are legible and have good quality. However, I encourage Authors to unify the style of figures with more than one element by using (a), (b),… both in the illustrations and figure captions. Marking subfigures by ‘a.)’ does not look well. Figures without marks (using ‘upper’, ‘lower’, etc., like figure 1) should also be equipped with (a), (b),… The quality of graph representing proposed processing algorithm (Figure 11) is rather low (the greater font size would improve the legibility). In general, the font size and style in figures should be unified throughout the whole manuscript (e.g., axis labels are sometimes blurred or too small).
10) There is an inconsistency between capture of Figure 5 and the text in lines 164-166. The text says that low-frequency information is related to lower scales, whereas scale 1 is connected with high frequency in the figure caption.
11) Figure 6 should be more explained in the main text, instead of discussing it in the caption.
12) Results. Where does the presented data come from, what is the source, was any specific object examined?
13) Conclusions. The conclusions appropriately support the presented results. If it is possible, I recommend to mention about any future plans concerning the topic of the current article. The fact that the undertaken problem can be further analysed could enhance the overall reception of the paper.
14) References. The formatting of references in the reference list should be revised to meet the rules presented in Journal’s Instructions for Authors and in the manuscript template (e.g., order of elements, using italic for journals’ named and bold for year). The citation formatting should also be revised, e.g., it should be [7,8] instead of [7], [8] in line 77; or [12–16] instead of [12]-[16] in line 80.
Round 2
Reviewer 2 Report
The paper revision does not include any ground truth, hence the authors have not provided any evidence to substantiate their proposition. Relying on hearsay information that the provider of the data is experienced in GPR does not change the fact that the paper doesn't provide evidence to support the work. The authors need to obtain data from core samples and measurements of moisture content to justify their work and include that in the paper. When that is done the paper will have some validity.
The authors have lost sight of basic scientific principles in that propositions regarding theory or processing algorithms must be supported by adequate experimental measurements and the quality and quantity of the latter should be understood and described.
Author Response
The content of this paper is not to compare GPR signals and real ground structures in the rail track area. Our aim is to characterize the fine signal details in radargrams, which are not visually accessible to a human observer. The potential of this pixel-wise segmentation will of course be analysed in detail in future experimental work, e.g. by detailed comparison with data from bore holes or trial pits.
Reviewer 4 Report
The manuscript has been significantly enhanced. Most of the remarks have been fully addressed. However, I would like to make some other, minor comments that should be considered by the Authors before acceptance.
MINOR COMMENTS
1) References. The references in the reference list are still not in full agreement with the rules presented in Journal’s Instructions for Authors and in the manuscript template (e.g., initials after surname, abbreviated journal name). Please correct. It can be beneficial to use automated reference managers, like Mendeley, in which the specific journal’s citation style can be applied.
2) Equations. The font was appropriately changed, however, the variables in equations should be written in italic throughout the whole manuscript.
Author Response
Thank your for your response.
- We have corrected the reference list style according to the MDPI manuscript template.
- We have changed variables in equations to italic font.